# Inter-Laboratory Comparison of RT-PCR-Based Methods for the Detection of Tomato Brown Rugose Fruit Virus on Tomato

**DOI:** 10.3390/pathogens11020207

**Published:** 2022-02-03

**Authors:** Marta Luigi, Ariana Manglli, Antonio Tiberini, Sabrina Bertin, Luca Ferretti, Anna Taglienti, Francesco Faggioli, Laura Tomassoli

**Affiliations:** Research Centre for Plant Protection and Certification, Council for Agricultural Research and Economics (CREA-DC), Via C.G. Bertero 22, 00156 Rome, Italy; marta.luigi@crea.gov.it (M.L.); ariana.manglli@crea.gov.it (A.M.); antonio.tiberini@crea.gov.it (A.T.); sabrina.bertin@crea.gov.it (S.B.); luca.ferretti@crea.gov.it (L.F.); anna.taglienti@crea.gov.it (A.T.); laura.tomassoli@crea.gov.it (L.T.)

**Keywords:** ToBRFV, conventional RT-PCR, real-time RT-PCR, validation, performance criteria, TPS

## Abstract

In 2020, a test performance study (TPS) for the specific detection of tomato brown rugose fruit virus (ToBRFV) was organized in the frame of the H2020 Valitest project. Since no validated tests were available, all the protocols reported in the literature were at first screened, performing preliminary studies in accordance with the EPPO standard PM 7/98 (4). Five molecular tests, two conventional RT-PCR and three real-time RT-PCR were found to be suitable and were included in the TPS. Thirty-four laboratories from 18 countries worldwide took part in TPS, receiving a panel of 22 blind samples. The panel consisted of sap belonging to symptomatic or asymptomatic leaves of *Solanum lycopersicum* and *Capsicum annuum*. The results returned by each laboratory were analyzed and diagnostic parameters were assessed for each test: reproducibility, repeatability, analytical sensitivity, diagnostic sensitivity and diagnostic specificity. All the evaluated tests resulted in being reliable in detecting ToBRFV and were included in an EPPO Standard PM 7/146—Diagnostics.

## 1. Introduction 

Tomato brown rugose fruit virus (ToBRFV) (genus *Tobamovirus*, family *Virgaviridae*) has a single-stranded positive sense RNA genome located in rigid elongated particles. It first emerged in Israel and Jordan in 2014 and 2015, respectively [1,2], and later on was reported in several European countries as well as in Central and North America, where it was first reported in Mexico in Autumn 2018 [3], and then it spread in California [4]. In 2019, ToBRFV was also reported in the Asian continent, in Turkey [5] and China [6] (for further information see EPPO global database distribution at https://gd.eppo.int/taxon/TOBRFV/distribution, accessed on 10 December 2021). ToBRFV is included in the EPPO A2 list and, since November 2019, has been regulated (Commission implementing regulation (EU) 2020/1191 in August 2020, and subsequent amendments, repealing Commission Implementing Decision (EU) 2019/1615) to prevent it spreading in the European Union. This regulation includes requirements for either plantlet or seed sampling and testing.

A prompt adoption of regulations was due to some biological features that favor a rapid spread of the tobamoviruses. Specifically, ToBRFV is transmitted through seed (contaminated seed coats) with a variable transmission rate (0.08–1.8%) [7,8], and by contact through human activities during crop production; furthermore, bumblebees’ transmission was reported [9]. Moreover, ToBRFV virions can survive for a long time in infected plant residues, contaminated soil, on tools and worker clothes, irrigation systems and greenhouse structures (such as poles, nets, pallets, etc.) contaminated with raw sap from infected plants [10]. For these reasons, the probability of further entry and the establishment of ToBRFV in the EPPO region is reported to be high, with a low uncertainty [11].

ToBRFV is known to infect *Solanum lycopersicum* (tomato) and *Capsicum annuum* (pepper) [3,12] and it can cause from a mild to a severe mosaic [2] as well as narrowing and necrosis on leaves [13]. Fruits can mainly show discoloration or marbling [6] and malformation [12], and, more rarely, brown rugosity (which gave the name to the virus) [2]. Seedlings for transplanting are generally asymptomatic.

Early detection and prompt adoption of effective phytosanitary measures are crucial steps to reduce the risk of entry and the spread of plant pests and to limit their damage. Regarding the detection, the performance evaluation and validation of diagnostic tests is essential to select and make available the most effective and reliable methods to be used in official controls. In the frame of the H2020 VALITEST project (an EU funded research project—https://www.valitest.eu, accessed on 10 December 2021) aimed at producing validation data for tests for which no or limited data are currently available, a test performance study (TPS) for molecular detection of ToBRFV in tomato and pepper leaves and fruits was organized and the results are reported. The tests included in the TPS were then reported in an EPPO standard (PM 7/146) for the diagnosis of ToBRFV [14].

## 2. Results

### 2.1. Intra-Laboratory Evaluation

Due to the lack of some validation data and the difficulty to compare those already available, preliminary trials were performed in-house to select the most suitable tests to be included in the TPS. According to the EPPO guidelines [15,16], analytical specificity (also referred to as inclusivity and exclusivity) and analytical sensitivity were first evaluated. Only those tests that showed 100% analytical specificity and a limit of dilution (LOD) ≤ 10^−1^ in the working conditions (Table 1), were included in the TPS. As reported in Table 1, five molecular tests, here referred to as ALK [12] and LOE [17], and three real-time RT-PCRs, here referred to as ISH [18], M&W [19] and PAN [20], fulfilled these requirements. All five tests were able to detect the target isolates, while none of them exhibited a cross reaction with other tobamoviruses. For the three real-time RT-PCR tests a similar level of analytical sensitivity was also recorded.

### 2.2. Test Performance Study

#### 2.2.1. Participants

The 34 participants in the TPS came from different parts of the world. The majority were from EU countries (77%), and the others were from Switzerland, Israel and New Zealand. Of the participants, 33% were in the VALITEST project consortium.

All the laboratories were able to submit their results, for a total of 150 data sets: 53 data sets for the two RT-PCRs (27 for ALK test and 26 for LOE test) and 97 data sets for the three real-time RT-PCRs (34 for ISH and M&W and 29 for PAN).

A remarkable number of deviations from the recommended protocols were recorded in the reports provided by participants. Regarding RNA extraction protocols, most laboratories used the suggested kit (81%), five laboratories used other kits (AccuPrepViral RNA Extraction kit, Bioneer; InnuPrep Plant RNA Kit, Analytik Jena; Maxwell RSC plant RNA, Promega; Plant RNA/DNA Purification kit, Norgen Biotek Corp.; TRIsure, BIOLINE) and one lab used the C-TAB extraction procedure [21]. Regarding the amplification, 3 out of 27 laboratories who performed the ALK test used amplification reagents different from those suggested by the organizer: two laboratories used other commercial kits (SS III/Platinum™ Taq Master Mix, Life Technologies and OneTaq^®^ One-Step RT-PCR Kit, NEB) and one a home-made master mix. Most of the deviations from the proposed protocols occurred in real-time RT-PCR where only 65% of the participants used the two recommended amplification kits (see the list of real-time RT-PCR kits and reagents used by the TPS participants in Appendix A).

#### 2.2.2. Data Set Evaluation

A set of 22 blind test items was used, composed of nine samples types (S1–S9) provided in duplicate or triplicate. Controls (negative isolation control—NIC, positive isolation control—PIC, positive amplification control—PAC and negative amplification control—NAC) were provided along with the sample panel. All the participants submitted results for all the controls (except one laboratory that did not test NAC), and none of the laboratories gave inconclusive results, for a total of 596 results for NIC, PIC, PAC and NAC (Table 2). Controls were used for a first evaluation of the effect of the deviations from the suggested real-time RT-PCR protocols, analyzing the average Cq values and the associated standard deviations, and for a preliminary quality check of the data sets. Thus, only data sets with all the concordant results on controls (Table 2) were deemed valid and considered for data analysis. The same analysis was considered not necessary for conventional RT-PCR due to the small number of participants that deviated from the proposed protocols.

According to the outliers’ results below reported in 2.2.4, some datasets were excluded from the analysis: the final number of valid data sets ranged from 71% to 81% for the different tests, for a total of 114 valid datasets (2508 samples).

#### 2.2.3. Repeatability and Reproducibility

Repeatability and reproducibility, defined as accordance (DA) and concordance (CO) [22], respectively, were calculated according to the parameters reported by Langton et al., 2002 [23]. For such evaluation, results in each valid data set were considered. Repeatability was evaluated for each test considering the results obtained by each repetition (Table 3).

Reproducibility was evaluated for each sample (Table 4) and considering the results obtained by each laboratory (see the table of reproducibility values obtained for all the tests by each participant in Appendix A). 

#### 2.2.4. Analytical Sensitivity

The samples from S3 to S7, prepared at different levels of dilution, were used to evaluate the overall analytical sensitivity (ASE). ASE was calculated modelling the results and establishing, when possible, the dilutions corresponding to a 50% or 95% probability of detection (Table 5; Figure 1). 

#### 2.2.5. Evaluation of the Other Performance Criteria

The performance of individual tests was also evaluated in terms of accuracy (ACC), diagnostic sensitivity (DSE) and diagnostic specificity (DSP), calculating the percentage of results that were inconclusive (INC), true negative (TN), false positive (FP), false negative (FN) and true positive (TP); for the real-time RT-PCR tests, the interpretation received from each laboratory (positive, negative or inconclusive) was taken into account. The results from the highly diluted samples (S3 and S4) were not considered because of their difficult interpretation. Inconclusive results were excluded in the calculation of ACC, DSE and DSP. As reported in Table 6, the test with the highest rate of inconclusive results was the ISH (4%). The accuracy of the tests ranged from 85 to 88%, diagnostic specificity from 86 to 98% and diagnostic sensitivity from 81 to 88%.

#### 2.2.6. Evaluation of the Deviations

As already reported, many laboratories deviated from the recommended protocol, especially in using different master mix reagents for real-time RT-PCR tests. A comparison of the accuracy values obtained by strictly following the suggested protocols vs the values obtained after deviations was carried out to evaluate the possible effect of these deviations on the performance of the test. For each test, the average percentage of accuracy for non-deviating protocols was always slightly higher than the percentage for deviating protocols but the differences were not significant (Student’s *t*-test *p* < 0.05) (Table 7; Figure 2).

## 3. Discussion

The intended scope of the TPS was to assess the performance criteria of the tests collected from different sources and laboratory experience at the time of the TPS to specifically detect ToBRFV in leaves, and to ascertain if they can be used in detection of the virus in the leaf and fruit of tomato and pepper. A preliminary study was conducted to select the most suitable diagnostic tests according to the intended scope. Firstly, all the tests were harmonized and standardized, employing selected one-step master mixes for amplifications and a commercial kit for total RNA extraction to reduce the risk of errors and process variability in performing the analysis. The tests were evaluated for their analytical specificity and only those exhibiting 100% of specificity were then evaluated for their analytical sensitivity. In the fixed working conditions, five molecular tests, two conventional RT-PCRs [12,17] and three real-time RT-PCRs [18,19,20] met the requirements in terms of performance criteria. All these tests showed 100% of inclusivity (in analyzing different target isolates) and exclusivity (in analyzing non-target viruses). The three real-time RT-PCR tests showed similar analytical sensitivity values in detecting ToBRFV, both in tomato and pepper leaves extracts, and, as expected, these values were higher than those obtained by the two conventional RT-PCR tests. Based on these results, the five molecular tests were included in the TPS and evaluated on a panel of 22 blind samples and 4 controls (NIC, PIC, PAC and NAC). Samples were tested for their homogeneity and stability before being shipped to the 34 laboratories registered for the TPS. Most of the participants were outside of the VALITEST project consortium and, as they granted a great expertise in testing pathogens such as ToBRFV and in performing molecular tests, decided to deviate from the recommended protocols, employing amplification reagents different from those suggested. In any case, no differences were noticed analyzing the results obtained from the controls and comparing the Cqs from PAC and PIC, thus indicating that the deviations did not affect the obtained results. 

The main performance criteria obtained for all the valid data set during the TPS are summarized in Figure 3. All the tests and all the participants (considering valid data sets) obtained acceptable results in reproducibility and repeatability; the lowest values of such performance criteria occurred only in those samples prepared at high dilution level (≤LOD of the tests). A decrease in sensitivity was noticed by comparing the values obtained in the TPS on the samples at different levels of dilution with the values obtained in the preliminary tests, especially for real-time RT-PCR tests. These differences likely accounted for the broader conditions in which TPS results are collected compared to an in-house validation, but, in any case, they were acceptable for a diagnostic protocol.

The good TPS results highlighted the reliability of the evaluated tests and the ability of the participants to correctly interpret the results. This consideration was also confirmed by the small number of inconclusive results reported by the participants. The numbers of true positive or true negative results were close to those expected and all the tests showed an acceptable accuracy value ranging from 85% to 88%. 

Among the conventional RT-PCR tests, the LOE and ALK tests showed the highest DSE and DSP, respectively. Among the real-time RT-PCR tests, the PAN test had the highest DSP, whereas ISH and M&W showed the highest DSE. The most reliable test is a tradeoff between these two parameters even if more often than not the obtained case is a test with a very high DSE that frequently lacks in DSP and the other way round. The three real-time tests resulted in also being robust since modification of the nucleic acid extraction procedures and/or master mix reagents did not affect their accuracy. 

In conclusion, considering the high number of laboratories who submitted results, all the tests provided satisfactory results in all the performance criteria evaluated. The only aspects that could affect the validity of the tests were: the high risk of contamination due to the handling of tobamoviruses, and the right assessment of a cut-off cycle in some conditions that is critical to better discriminate specific reactions from not-specific cross reactions.

## 4. Materials and Methods

A wide web search was carried out to collect data on the available methods for ToBRFV detection. Since at the time of the TPS only few tests had validation data with a low comparability, the work was divided in two steps: (i) intra-laboratory pre-tests to evaluate the feasibility of the protocols in accordance with the TPS scope; (ii) TPS organization, including the selection of the participants, preparation of the test items, shipping of samples and reagents, analysis of the results following the EPPO Standard 7/122 [22].

### 4.1. Sample Collection and Total RNA Extraction

Samples used in both the pre-tests and TPS included isolates of tobamoviruses belonging to CREA-DC collection or purchased at DSMZ company (Leibniz Institute, Braunschweg, Germany). The following ToBRFV isolates, available at the first working step, have been used as target reference material for the evaluation of inclusivity: ToB-SIC21/19; ToB-SIC22/19; ToB-SIC23/19; ToB-SIC24/19; ToB-SIC 25/19; ToB-PIE105/2019 (originally isolated from *S. lycopersicum* belonging to the CREA-DC collection), and ToBRFV PV-1236, PV-1241 and PV-1244 (obtained as freeze-dried leaf materials from the DSMZ collection). Isolates of other tobamoviruses from the DSMZ collection were used as non-target items (tomato mosaic virus—ToMV PV-0141; tobacco mosaic virus—TMV PV-1252; pepper mild mottle virus—PMMoV PV-0165; bell pepper mottle virus—BPeMV PV-0170; tomato mild green mottle virus—TMGMV PV-0124). Healthy plants of tomato and pepper were used for providing negative samples and NIC in the panel. Leaves and all the fruits of tomato (healthy or infected) were ground in PO_4_ buffer 0.1 M pH 7.2, in concentration 1:10 w/v for leaves and 1:20 w/v for fruits. The obtained sap was used either fresh or freeze-dried. In total, 100 µL of the sap (fresh or rehydrated with RNase free water) were added to 380 µL of Lysis Buffer of RNeasy Plant Mini kit (Qiagen, Hilden, Germany), and RNA was extracted according to the manufacturer’s instructions.

### 4.2. Conventional RT-PCR Amplification

Conventional RT-PCR tests published by Alkowni et al.—ALK [12], and Rodriguez-Mendoza et al., [17] were harmonized using One-Step RT-PCR kit (Qiagen Sciences, Germantown, MD, USA), according to the manufacturer’s instructions. The annealing temperature and the primer concentration were adjusted following the original publications (Table 1). Primer pair from Rodriguez-Mendoza et al. [17] has been included in a ready-to-use kit developed by Loewe (tomato brown rugose fruit virus—Complete One-Step Reverse transcriptase PCR Reaction Kit, (Loewe Biochemica GmbH, Sauerlach, Germany). After the preliminary study, this primer pair was used only within the Loewe kit, according to the manufacturer’s instructions (LOE). In the preliminary tests, all the PCRs were conducted in a C1000 Touch thermal cycler (Bio-Rad). All amplified products were analysed by electrophoresis in 1.2% agarose gel and stained with ethidium bromide. 

### 4.3. Real-Time RT-PCR

Real-time RT-PCR tests published by Ishi-Veg—ISH [18]; Menzel and Winter—M&W [19] and Panno et al.—PAN [20] were harmonized using TaqMan^®^ RNA-to-Ct™ 1-Step Kit (Life Technologies, Carlsbad, CA, USA) and iTaq™ One-Step RT-PCR Kit for Probes (Bio-Rad, Hercules, CA, USA). Generally, the primers and probe concentrations were maintained as reported in the original publication (Table 1). The concentrations in the ISH protocol were modified (Table 1) to minimize no-specific cross reactions that occurred with high Ct without affecting the performance of the test. In the preliminary tests, all the amplification reactions were conducted in a CFX96 optical reaction module with C1000 Touch thermal cycler (Bio-Rad).

### 4.4. In-House Validation

Molecular tests were selected according to their analytical specificity (inclusivity and exclusivity) and analytical sensitivity evaluated as suggested by [15,16]. Sap samples from 3 tomato (ToB-SIC21/19, ToB-SIC22/19 and ToB-SIC23/19) and from 3 pepper infected plants (artificially inoculated with ToB-SIC24/19; ToB-SIC 25/19; ToB-PIE105/2019) were 10-fold serially diluted in healthy tomato or pepper leaf sap up to 10^−8^, the last dilution providing the limit of dilution (LOD) used for assessing the analytical sensitivity. The analytical specificity was evaluated by comparing in silico primers and probes sequences with the sequences from genomic library (by Blast tool in NCBI) and then testing target isolates (inclusivity) and no-target isolates (exclusivity), reported in Table 1.

### 4.5. TPS Participants

Criteria for the selection of the participants were defined by the project organizer (according to the VALITEST deliverable D1.1 “Minimum performance parameters to select tests for validation and selection of laboratories for TPS”). Thirty-four laboratories from 19 European and non-European countries (Figure 4) took part to the TPS and all of them were able to submit their results (Appendix A).

### 4.6. Panel of Test Items’ Composition and Preparation of the Shipping

Thirty-four identical panels of samples were prepared, each including 22 blind test items and 4 controls (NIC, PIC, PAC and NAC—water control). The 22 test items included 9 samples types, composed as follows: two negative samples in duplicate (one tomato and one pepper for a total of 4 samples); one positive sample infected at low concentration and one infected at medium concentration, both in duplicate; a third positive sample was sent at different level of dilutions in duplicate or triplicate (Table 8).

The nine samples, the NIC and the PIC consisted of 0.5 mL of freeze-dried sap obtained from leaves (NIC) or fruits (PIC) and stored at room temperature before shipping, the PAC consisted of a total RNA sample extracted as reported above and kept at −20 °C until shipping and sent in dry ice. Ready-to-use mixtures of primers for conventional RT-PCR and primers and probes for all the real-time RT-PCR tests were included in the shipping. Samples and reagent mixtures were tested for their homogeneity and stability before the shipping according to the EPPO standard 7/122 [22]. Homogeneity was ascertained testing 9 aliquots of each sample randomly chosen from the prepared batch, using M&W test after a week from the preparation; PAC was tested in three technical repetitions. Stability was ascertained retesting randomly chosen aliquots after 20 weeks (after receiving all the results from laboratories). Stability of total RNA was carried out under conditions that mimicked transport (storing total RNA aliquots 3 days at room temperature, approximately 22 °C).

### 4.7. Evaluation of the Performance Criteria

Performance criteria and validation procedures were established following guidelines from the EPPO standards PM 7/98 [16] and PM 7/122 [22]; repeatability and reproducibility were calculated applying the method from Langton et al. [23]. Analytical sensitivity was evaluated for each test, data of the diluted samples were used to adjust binomial generalized linear models (bGLM) with logit link between the dilution (expressed by the base 10 negative exponent of the corresponding dilution) and the detection outcome. The dilution was made using healthy sap prior to RNA extraction ranging from 10^−8^ to 10^−2^.

### 4.8. Outliers Results

Data sets were considered outliers, and excluded from analysis, if: (a) results of controls were non-concordant; (b) accuracy statistically different from the average of accuracy obtained by all laboratories; (c) results of one test were incomplete (e.g., no technical repetition reported); (d) the number of undetermined results was significantly different from the other laboratories (n. undetermined/inconclusive > average undetermined + 3 σ).

## Figures and Tables

**Figure 1 pathogens-11-00207-f001:**
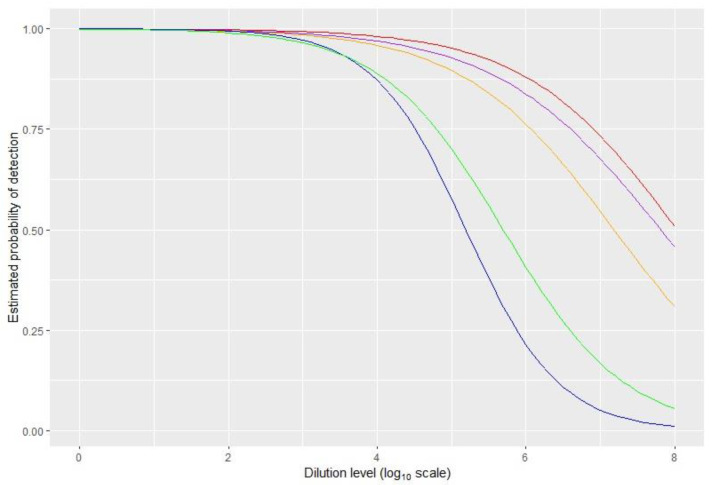
Analysis of the probability of detection (POD) for all the tests using bGLM modelling. The best fitting curves are plotted for all the tests (blue—ALK; green—LOE; purple—ISH; red—M&W; orange—PAN).

**Figure 2 pathogens-11-00207-f002:**
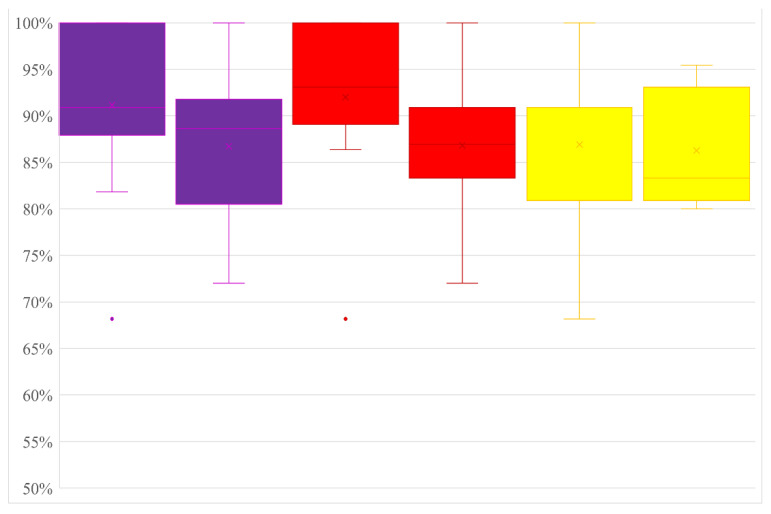
Box plot of the accuracy values obtained for each test with not-deviated (left) and deviated (right) protocols used by all the laboratories (purple ISH, red M&W and yellow PAN).

**Figure 3 pathogens-11-00207-f003:**
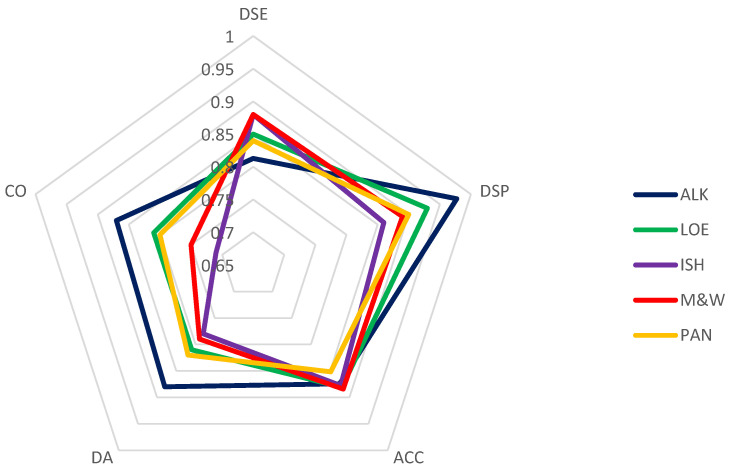
Summary of the main criteria obtained for each protocol tested in the TPS: diagnostic sensitivity (DSE); diagnostic specificity (DSP); accuracy (ACC); repeatability (DA); reproducibility (CO).

**Figure 4 pathogens-11-00207-f004:**
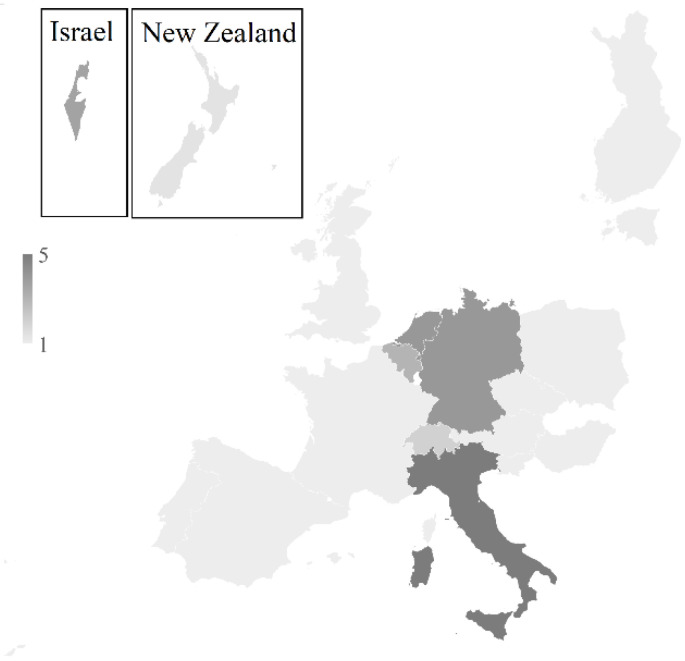
Geographical origin of TPS participants.

**Table 1 pathogens-11-00207-t001:** Performances of the tests selected for the TPS after intra-laboratory trials, and the most relevant working conditions.

		Conventional RT-PCR		Real-time RT-PCR		
**Test**		**ALK**	LOE	ISH	M&W	PAN
Ref.		[12]	[17]	[18]	[19]	[20]
Conditions	Primers and/or probes final concentration (each)	0.2 µM primer	-	0.1 µM probe 0.15 µM primer	0.3 µM probe 0.25 µM primer	0.5 µM probe 0.25 µM primer
	T annealing	58 °C 30″	55 °C 20″	60 °C 1′	60 °C 1′	60 °C 1′
Analytical specificity	Inclusivity	100%	100%	100%	100%	100%
	Exclusivity	100%	100%	100%	100%	100%
Analytical sensitivity	Tomato	10^−3^	10^−5^	10^−7^	10^−7^	10^−7^
	Pepper	10^−1^	10^−3^	10^−3^	10^−3^	10^−3^

**Table 2 pathogens-11-00207-t002:** Number of concordant/non-concordant results obtained from the controls by all laboratories considered both conventional and real-time tests and average Cq values (±Std. Dev.) recorded for each control with the different real-time RT-PCR protocols.

		NIC	PIC	PAC	NAC
Results	Concordant (%)	131 (87%)	145 (97%)	146 (97%)	141 (94%)
	Non-concordant (%)	19 (13%)	5 (3%)	4 (3%)	5 (3%)
	Untested (%)	0	0	0	4 (3%)
	ISH	36.28 ± 3.27	15.32 ± 3.05	19.94 ± 2.75	38.72 ± 3.02
	M&W	36.53 ± 3.17	15.68 ± 2.65	20.28 ± 3.43	39.23 ± 2.04
	PAN	36.81 ± 3.10	18.05 ± 3.79	22.58 ± 2.15	39.92 ± 0.37

**Table 3 pathogens-11-00207-t003:** Repeatability values obtained for S1–S9 samples with the different tests.

	S1	S2	S3	S4	S5	S6	S7	S8	S9	Total
**ALK**	100%	95%	97%	72%	76%	100%	100%	91%	61%	**88%**
**LOE**	95%	79%	91%	50%	75%	100%	100%	91%	49%	**81%**
**ISH**	80%	71%	50%	72%	100%	100%	100%	54%	75%	**78%**
**M&W**	88%	73%	49%	71%	100%	100%	100%	53%	78%	**79%**
**PAN**	95%	70%	63%	65%	100%	100%	90%	72%	79%	**82%**

**Table 4 pathogens-11-00207-t004:** Reproducibility values obtained for S1–S9 samples from the different tests.

	S1	S2	S3	S4	S5	S6	S7	S8	S9	Total
**ALK**	100%	95%	97%	69%	76%	100%	100%	91%	57%	**87%**
**LOE**	95%	79%	91%	47%	75%	100%	100%	91%	49%	**81%**
**ISH**	79%	71%	49%	63%	88%	88%	96%	54%	66%	**73%**
**M&W**	88%	73%	48%	65%	92%	92%	100%	52%	72%	**76%**
**PAN**	95%	70%	62%	65%	100%	100%	90%	70%	79%	**81%**

**Table 5 pathogens-11-00207-t005:** Detection limit at 50% and 95% calculated for all the tests.

	ALK	LOE	ISH	M&W	PAN
Log dilution factor at 50%	5.2	5.7	7.8	N.A.	7.2
Log dilution factor at 95%	3.4	3.3	4.6	5	4.2

**Table 6 pathogens-11-00207-t006:** Rate of true negative (TN), true positive (TP), false negative (FN) and false positive (FP) results obtained with the different tests and corresponding to the values of accuracy (ACC), diagnostic sensitivity (DSE) and diagnostic specificity (DSP). NS = not significant at *p* < 0.05.

		ALK	LOE	ISH	M&W	PAN
**N. valid data set**		22	21	24	25	22
**N. of samples**		352	336	384	389	344
**TN (%)**	(TN/N^−^%)	129 (98%)	118 (94%)	79 (82%)	88 (88%)	78 (89%)
**TP (%)**	(TP/N^+^%)	178 (81%)	178 (85%)	224 (85%)	256 (85%)	215 (81%)
**FN (%)**	(FN/N^+^%)	41 (19%)	32 (15%)	32 (11%)	34 (11%)	42 (16%)
**FP (%)**	(FP/N^−^%)	3 (2%)	8 (6%)	13 (14%)	11 (11%)	9 (10%)
**INC (%)**	(INC/N%)	1 (0%)	0 (0%)	16 (4%)	11 (3%)	8 (2%)
**ACC**	(TP+TN)/ (TP+TN+FP+FN)	87%	88%	88%	88%	85%
**CI_ACC_ 95%**		66–100%	73–100%	66–100%	66–100%	54–94%
***p*-Value Fisher ACC**		NS	NS	NS	NS	NS
**DSE**	TP/TP+FN	81%	85%	88%	88%	84%
**CI_DSE_ 95%**		43–100%	57–100%	59–100%	56–100%	48–100%
***p*-Value Fisher DSE**		NS	NS	NS	NS	NS
**DSP**	TN/TN+FP	98%	93%	86%	89%	90%
**CI_DSP_ 95 %**		95–100%	88–99%	80–92%	79–99%	74–100%
***p*-Value Fisher DSP**		NS	NS	NS	NS	NS

**Table 7 pathogens-11-00207-t007:** Average, minimum and maximum accuracy values obtained from laboratories that did not make deviation from the recommended protocols (no-dev) and from those that deviated (dev).

	ISH		M&W		PAN	
	no-dev	dev	no-dev	dev	no-dev	dev
**N. valid dataset**	14	10	14	11	13	10
**Acc %** **Av**	91%	87%	92%	87%	87%	84%
**Acc % Min**	68%	72%	68%	72%	68%	68%
**Acc % Max**	100%	100%	100%	100%	100%	95%

**Table 8 pathogens-11-00207-t008:** Number and characteristics of sample types used in the TPS and the expected outcomes. * Results ≤ LOD.

SAMPLE TYPE	SAMPLE ID	ISOLATE	HOST	SANITARY STATUS	EXPECTED OUTCOME (RT-PCR)	EXPECTED OUTCOME (REAL-TIME)
**S1**	ToBRFV-M-1; M-2	-	*S. lycopersicum*	Healthy	Negative	Negative
**S2**	ToBRFV-M-3; M-4	-	*C. annuum*	Healthy	Negative	Negative
**S3**	ToBRFV-M-5; M-6; M-7	ToB-SIC 21/19	*S. lycopersicum*	10^−8^	Negative	Positive *
**S4**	ToBRFV-M-8; M-9; M-10			10^−6^	Positive *	Positive
**S5**	ToBRFV-M-11; M-12; M-13			10^−4^	Positive *	Positive
**S6**	ToBRFV-M-14; M-15; M-16			10^−2^	Positive	Positive
**S7**	ToBRFV-M-17; M-18			10^0^	Positive	Positive
**S8**	ToBRFV-M-19; M-20	ToB-SIC 23/19	*S. lycopersicum*	Low (10^−6^)	Positive *	Positive
**S9**	ToBRFV-M-21; M-22	ToB-SIC 25/19	*S. lycopersicum*	Medium (10^−4^)	Positive *	Positive
**NIC**	ToBRFV-M-NIC	- ToB-SIC 24/19	*S. lycopersicum* *S. lycopersicum*	10^0^	Negative	Negative
**PIC**	ToBRFV-M-PIC			10^0^	Positive	Positive
**PAC**	ToBRFV-M-PAC	ToB-SIC 22/19	*S. lycopersicum*	10^−2^	Positive	Positive

## Data Availability

The data presented in this study are available on request from the corresponding author. The data are not publicly available due to the confidentially agreement signed by the participants and TPS organizer.

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
