# Peer review of "Inter-Laboratory Comparison of RT-PCR-Based Methods for the Detection of Tomato Brown Rugose Fruit Virus on Tomato"

_pathogens, 2022, doi:10.3390/pathogens11020207_

Round 1
Reviewer 1 Report
Tomato brown rugose fruit virus (ToBRFV) is so important pathogens for tomato production and seed trade worldwide now. This validation of diagnostic tests are useful infromation for detection for ToBRFV.
Author Response
We thank you the reviewer for appreciating our manuscript. We found no comments or request of clarifications in the review, therefore we think there are no answers to give.
Reviewer 2 Report
In the last few years,tomato brown rugose 9 fruit virus (ToBRFV) spreads quickly across the world and has caused severe threat to the world production of tomato and pepper. Therefore, establishment of efficient diagnostic methods for the detection of ToBRFV in tomato and pepper leaves is urgent and would be very useful for the prevention and control of this viral disease. In this work, the author assessed and compared the efficacy of conventional RT-PCR and three real-time RT-PCR for the detection of ToBRFV by analyzing the data from a number of laboratories from several countries. They proposed that the RT-qPCR methods showed the best performance. Overall, this work provided important reference for the diagnosis of ToBRFV in plant leaves.
Author Response

(The authors gave the same response as above.)

Reviewer 3 Report
Commercial detection kits developed by biotech companies are useful for the detection and prevention of pathogens in plant crops. However, the efficacy and reliability of the test need to be validated and standardized in practice. In this paper, the authors evaluated the performance of the detection test on a specific virus in tomatoes and peppers. This could be a useful example for the evaluation of other detection methods. I have some comments to improve the manuscript:
- Could you define the 1x dilution of samples? Could you explain why limit dilution below 0.1?
- How much template (in nanogram or microgram) was used in each PCR reaction
- Real-time RT-PCR will provide quantitative results if there is a standard curve. Did you obtain the value of virus titers so that we can quantify the intensity of virus infection in the samples? Otherwise, Did you use any other ways to obtain quantitative results from real-time RT-PCR
- The specificity of PCR decides whether the signal from detection is from real-pathogen or nonspecific signals. We can not evaluate just based on the sequence of primers provided by the test. This need real data from sequencing the PCR products and blasting them to the database? How accurate of the signals
Author Response
We thank the reviewer for giving us the opportunity to better clarify the purpose of the manuscript and thus improve it. Specifically, it is a comparison of diagnostic tests already published and developed by the authors and compared by us using standard parameters provided by international technical standards (EPPO, IPCC). In particular, the comparison of the tests was performed always using the same set of samples from in vivo infections, therefore without any quantification of the target. Specifically, total RNA was extracted from ToBRFV-infected tomato and pepper leaves, that for dilution tests, it was diluted with RNA extracted from uninfected tomato and pepper leaves respectively. The standard curve of the real time RT-PCRs and the nature of the amplicons obtained with the end point RT-PCRs had already been verified by the authors who had developed and set up the methods. That said, below are our answers:
- Could you define the 1x dilution of samples? Could you explain why limit dilution below 0.1?
Samples at 1x dilution are never reported in the manuscript, probably the reviewer intends the samples reported as 100, that is “undiluted samples”. It means that the total RNA from these samples were obtained directly from infected field samples. Also limit dilution below 0.1 is never reported, probably the reviewer intends the 10-1 that corresponds to a ten fold dilution of the undiluted (100) samples. The dilution in base 10 (tenfold) is the classic formula used to establish the limit of dilution (LoD) for viral agents.
- How much template (in nanogram or microgram) was used in each PCR reaction
As previously mentioned, the aim of the work was not to evaluate the performance of the single tests, but to compare them with the same set of samples. For this reason we did not consider necessary to quantify the ng or µg of target, taking into account that the target was represented by total RNA (therefore including the target viral RNA and the plant RNA). Having used the exact same extracts for all tests, the purpose of comparing the different tests was achieve as requested by the international standard in performing a test performance study (TPS)
- Real-time RT-PCR will provide quantitative results if there is a standard curve. Did you obtain the value of virus titers so that we can quantify the intensity of virus infection in the samples? Otherwise, Did you use any other ways to obtain quantitative results from real-time RT-PCR.
The performance of the real time RT-PCRs were evaluated by the authors that developed them in order to give also the quantitative results and standard curve of the method. Our aim was not to verify the quantitation but the sensibility and specificity of the tests using the same set of samples.
- The specificity of PCR decides whether the signal from detection is from real-pathogen or nonspecific signals. We can not evaluate just based on the sequence of primers provided by the test. This need real data from sequencing the PCR products and blasting them to the database? How accurate of the signals
The answer is the same of above. The authors that developed and set up the end point RT-PCRs had already verified the nature of the products by sequencing them. In this case we also sequenced the products obtained from the end point RT-PCRs but only in order to verify the nature of our infected samples. We did not report the sequencing results in the paper since, in our opinion, not necessary.
Reviewer 4 Report
Please find comments on the manuscript inserted in attached file.

Author Response
We thank the reviewer for highlighting some critical aspects of the manuscript. We agreed on all the suggestions and modified the paper. We have submitted the revised manuscript in track changes format. Below you can see our responses to the comments of the reviewer:
- We agree and modified according the suggestion
- We agree and modified according the suggestion
- We agree and modified according the suggestion
- We agree and modified according the suggestion
- a.
- We agree and included the reference
- We agree and modified according the suggestion
- The results were already included we specified better in the text.
- a.
- According to the reviewer comment we understood that probably the text is not clear enough. Data from conventional RT-PCR were not included in analysing the deviation because only few laboratories deviated from the proposed protocols. In particular three participants deviated from the ALK and only two from the LOE. Due to the small number of deviations occurred we think that this analysis was not necessary for convention RT-PCR. We amended the text to better clarify this point.
- We agree and modified taking in account also the comment number 2
- We agree and modified according the suggestion
- We agree and added the required information
- We used all the fruits to create the sap that we sent to the TPS participants. We amended the text
- We added the required information
- We agree and modified according the suggestion
- We agree and modified according the suggestion
- We agree and modified according the suggestion
- Due to the high titre of ToBRFV in plants (both symptomatic and asymptomatic) in our opinion this information could not be useful
- We agree and modified the table
- We agree and modified the text
- We agree and modified the text
